Resource

# Intensive single-cell analysis reveals immune-cell diversity among healthy individuals

Yukie Kashima[1],*, Keiya Kaneko[1],*, Patrick Reteng[2],*, Nina Yoshitake[1], Lucky Ronald Runtuwene[3], Satoi Nagasawa[1,4] ORCID, Masaya Onishi[1], Masahide Seki[1], Ayako Suzuki[1], Sumio Sugano[5,6], Mamiko Sakata-Yanagimoto[7], Yumiko Imai[8], Kaori Nakayama-Hosoya[3], Ai Kawana-Tachikawa[3], Taketoshi Mizutani[1], Yutaka Suzuki[1] ORCID

**Immune responses are different between individuals and personal health histories and unique environmental conditions should collectively determine the present state of immune cells. However, the molecular systems underlying such heterogeneity remain elusive. Here, we conducted a systematic time-lapse single-cell analysis, using 171 single-cell libraries and 30 mass cytometry datasets intensively for seven healthy individuals. We found substantial diversity in immune-cell profiles between different individuals. These patterns showed daily fluctuations even within the same individual. Similar diversities were also observed for the T-cell and B-cell receptor repertoires. Detailed immune-cell profiles at healthy statuses should give essential background information to understand their immune responses, when the individual is exposed to various environmental conditions. To demonstrate this idea, we conducted the similar analysis for the same individuals on the vaccination of influenza and SARS-CoV-2. In fact, we detected distinct responses to vaccines between individuals, although key responses are common. Single-cell immune-cell profile data should make fundamental data resource to understand variable immune responses, which are unique to each individual.**

## Introduction

The human immune system consists of ingenious immune cells. It is widely known that the immune cells are collectively responsible for the versatile immune responses of an individual by shaping the immune landscape. The immune landscape should differ between individuals with distinct medical history and lifestyles, depending on genetic backgrounds and geographic origins. However, the influences and consequences of how immune cells maintain the previous memory of immune responses in a healthy state and respond to stimulation in normal life are largely unknown. It is partly because current knowledge on immune responses has been accumulated from laboratory animal models or diseased individuals. Even among individuals with a healthy appearance, infections, which may be mostly asymptomatic, occur daily.

Diverse immune responses, whether mild or severe, primarily occur at the infection site. However, it is commonly accepted that the immune profile is represented, at least in part, by circulating white blood cells, that is, peripheral blood mononuclear cells (PBMCs). Both innate immune cells make up the PBMCs. Including the innate immune cells are monocytes, which will mature into macrophage or dendritic cells (DCs) when they are recruited, and natural killer (NK) cells (Nicholson, 2016).

The adaptive immune responses depend on the specific recognition of an antigen by T cells or B cells at its recognized part called the epitope (Minervina et al, 2019). During the development of T and B cells, a process called VDJ recombination allows each cell to obtain a set of unique VDJ segments that determines the sequence of the antigen-binding site presented on the cell surface. Cells sharing the same VDJ sequence are said to have the same "clonotype." In T cells, the VDJ segments of the TCR, consisting of $\alpha$ and $\beta$ chains, are imperative to function properly. Similarly, presented on the outer membrane of B cells are B-cell receptors (BCRs), consisting of heavy and light chains of the immunoglobulin (IgH and IgK or IgL).

Conventionally, flow cytometry and fluorescence-activated cell sorting, or more recently bulk RNA-seq, have been the standard methods for monitoring the states of PBMCs and the immune systems they represent (Stubbington et al, 2017). However, substantial concerns have been raised about these methods. The main drawback is that, although these methods offer high-throughput and extensive gene detection, the obtained data would come from the cell mass in bulk; hence, the gene expression for a particular

[1]Department of Computational Biology and Medical Sciences, Graduate School of Frontier Sciences, The University of Tokyo, Kashiwa, Japan    [2]Division of Collaboration and Education, International Institute for Zoonosis Control, Hokkaido University, Sapporo, Japan    [3]AIDS Research Center, National Institute of Infectious Disease, Tokyo, Japan    [4]Division of Breast and Endocrine Surgery, Department of Surgery, St. Marianna University School of Medicine, Kawasaki, Japan    [5]Institute of Kashiwa-no-ha Omics Gate, Kashiwa, Japan    [6]Future Medicine Education and Research Organization at Chiba University, Chiba-city, Japan    [7]Department of Hematology, Faculty of Medicine, University of Tsukuba, Tsukuba, Japan    [8]Laboratory of Regulation for Intractable Infectious Diseases, National Institutes of Biomedical Innovation, Health and Nutrition (NIBIOHN), Osaka, Japan

Correspondence: ysuzuki@k.u-tokyo.ac.jp
*Yukie Kashima, Keiya Kaneko, and Patrick Reteng contributed equally to this work.

cell population is not represented separately. Also, it has been impossible to analyze the VDJ patterns, which are unique to individual cells, especially in association with the status of their expressing immune cells. Single-cell RNA-seq (scRNA-seq), a recently introduced method, enables a detailed observation at the single-cell level (Stubbington et al, 2017). With this single-cell analytical approach, cellular heterogeneity previously masked using bulk RNA-seq is now open to investigation to assess the response of a certain cell to an identified antigen.

Using scRNA-seq, recent studies have illustrated in great detail how the immune cells practically change their profiles in response to disease and infection. For example, a study on the PBMCs of patients with moderate to severe coronavirus disease 2019 (COVID-19) reported that the relative abundance of naive and activated T cells, mucosal-associated invariant T cells (MAIT), and monocyte-derived DCs decreased with disease severity, whereas T cells, plasma B cells, classical monocytes, and platelets increased (Zhang et al, 2020). Particularly, the timing and degree of induction of a subclass of T cells, called $\gamma \Delta$ T cells, appeared to be an important factor in determining the severity of the infections. Furthermore, it is noteworthy that all the cell populations except for the activated T cells were restored at convalescence.

In contrast to disease states, the extent to which immunity may differ among healthy individuals remains almost totally elusive. Although fundamental information on which various immune responses occur should be available (Carr et al, 2016; Lakshmikanth et al, 2020; Huang et al, 2021), only a handful of studies have been reported. TCR-VDJ gene-targeting PCR analysis revealed that TCR repertoires of memory T cells are at least partially specific to individuals. In contrast, TCRs from naive T cells showed no such individuality. The diverse immunological profiles, depending on individuals, may reflect their genetic background, infection history, and interactions with environments over a lifetime. However, how such diversity is acquired, maintained, and serves as a ground state for immune responses in generally healthy individuals remains almost totally unknown.

In this study, we describe our observations of the immune profile heterogeneity among healthy adults at the single-cell level, measured using scRNA-seq of the cellular transcriptomes and the VDJ repertoires. The profiles were further perturbed by vaccination against influenza viruses and severe acute respiratory syndrome coronavirus 2 (SARS-CoV-2). The resulting observations should be explained by the distinct "personal immunological landscape" shaped by each individual throughout their life and their present infection state, although participants reported a good health state during the study period. The broad aim of this study was to help understand baseline diversity in control groups regularly used in immunological studies of disease.

## Results

### Generation and evaluation of the scRNA-seq data for healthy individuals

To characterize the immunological landscape of seven healthy individuals (H1 to H7), their PBMCs were collected and used for the

following analyses. The overall study design, schematic illustration of sample collection, and processing are shown in Fig 1A. Refer to the Materials and Methods section for further details on the procedure. The personal information of the individuals is summarized in the inset table (Fig 1A, inset table).

A droplet-based scRNA-seq (Chromium of 10X Genomics) was performed for all samples. Particularly for H1 and H2, PBMCs were sampled at nine time points over months (Fig 1A and Table S1). On average, 240,207,958 reads were obtained for a single sample. An average of 28,826 reads were assigned for a single cell as the 5′-end mRNA gene expression information (Table S2). Each sample was individually clustered and visualized by UMAP (Figs 1B and S1A). The images were mostly overlapped between individual experiments (Fig S1A; for more details, see Table S2). Each cluster was annotated for a cell type using canonical cell markers (Figs 1B and C and S1B and C). The cells belonging to each cell type were counted as the corresponding cell populations. The relative percentage of major cell types constituting the PBMCs was calculated for all the datasets (for statistics, see Tables S3 and S4).

First, we evaluated the reproducibility and reliability of the data obtained. At the same time, the effect of sample freezing was also evaluated. It is convenient to freeze samples after collection and keep them in a freezer until an appropriate time for library preparation. However, the exact effect sample freezing has on the PBMC transcriptome has not been fully evaluated. For this purpose, we prepared libraries from the same material to subject to two conditions: fresh (H1 day 0 fresh and H2 day 0 fresh) and frozen (H1 day 0 frozen and H2 day 0 frozen) (Fig 1D, and red dotted line). Except for several specific particular cell types or a small group of genes in particular cell types that were excluded from the following analysis, there was no noticeable difference between the fresh and frozen sample in the total cell populations and gene expressions (Fig S2A and B; also, note that some NK cells seemed damaged by the sample freezing, which were detected as the increased proportion of mitochondria genes; other low correlated genes are shown in Table S5). Therefore, we used frozen samples for further analyses. In the following analyses, we will describe some characteristic features between different individuals (see below). However, these distinct features were within the range of daily changes; therefore, each datum was represented by nine independent experimental replicates. Collectively, we concluded that the collected data should be highly reproducible and reliable for further analysis.

### Diversity of the scRNA-seq profiles between different individuals and different time points

When we examined the resulting scRNA-seq profiles (Fig 1D and Table S4), the annotated cell type composition roughly agreed with those previously estimated (Verhoeckx et al, 2015), that is, typically, lymphocytes (T cells, B cells, and NK cells) ranging 70–90%, monocytes accounting for 10–20%, and DCs and other populations being rare. Within the lymphocyte population, cell types include CD3$^+$ T cells (70–85%), B cells (5–10%), and NK cells (5–20%). The CD3$^+$ T cells consist of CD4$^+$ T cells and CD8$^+$ T cells in approximately 2:1 ratio.

Despite the overall concordance with previous estimates, cell compositions differed across individuals and sampling time points

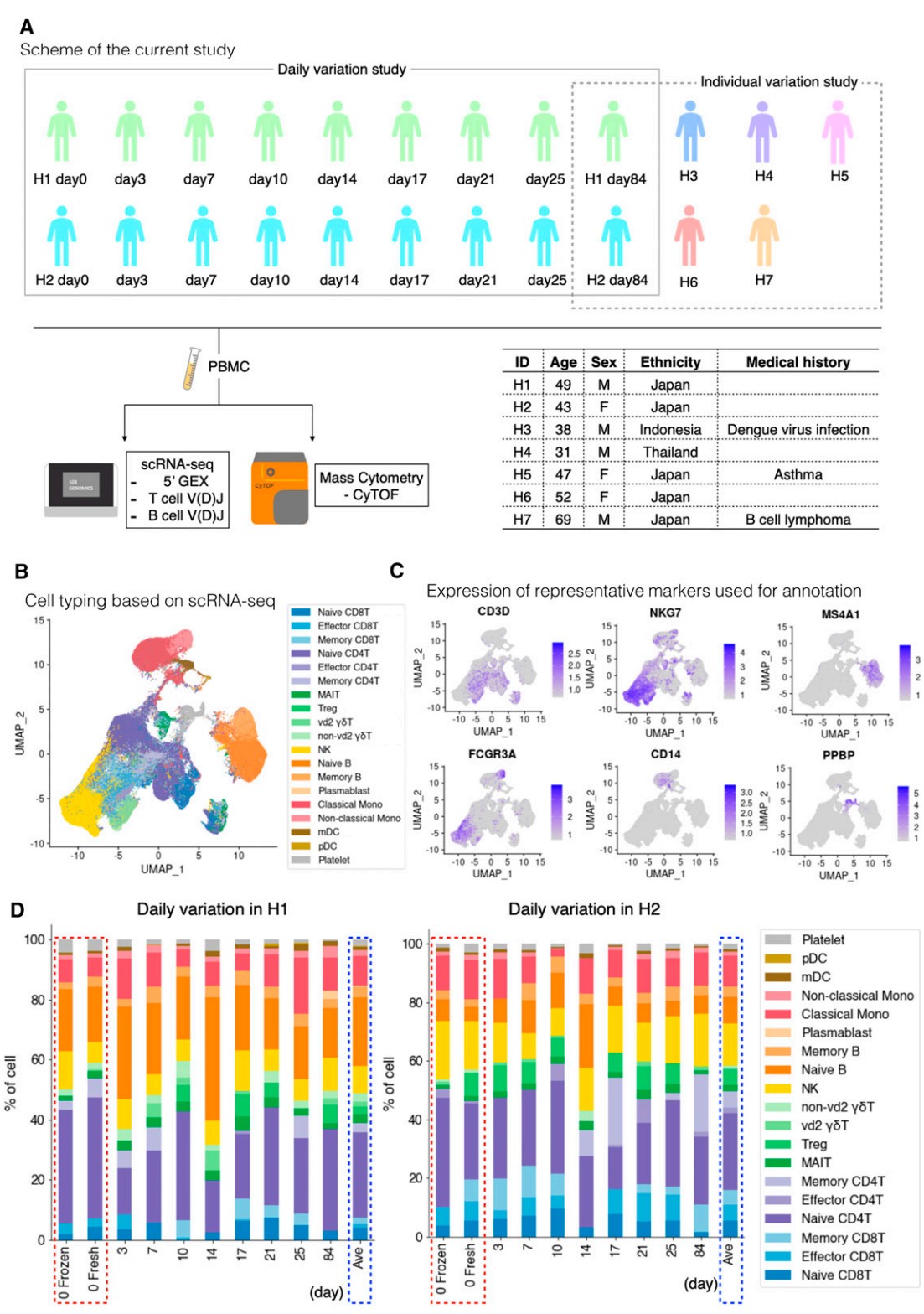

**Figure 1.  Characterization and evaluation of the scRNA-seq datasets.**
**(A)** Scheme and sample list of the present study. For the daily variation study, the multiple time point samples were collected from H1 and H2 at the shown time points. For the intraindividual variation study, PBMC samples were collected from seven participants, H1–H7. As illustrated, the collected samples were subjected to the transcriptome and proteome analyses (top). The medical history information about donors is shown in the table (bottom right). **(B)** Cell type annotation. UMAP plot showing clusters colored by cell types. **(C)** Expression of representative markers for cell annotation. Only the selected markers are shown. For others, see Fig S1B and C. We used the following markers: CD3D (T cell), NKG7 (NK cell), MS4A1 (B cell), CD14 and FCGR3A (monocyte), and PPBP (platelet). **(D)** Structure of PBMC at each time point of H1 (left) and H2 (right). The *x-axis* shows the day after first sampling, and the *y-axis* shows the percentage of each cell component. Bars with red dotted line show the data comparison of a fresh and frozen sample, and blue dotted line shows the average of each person.

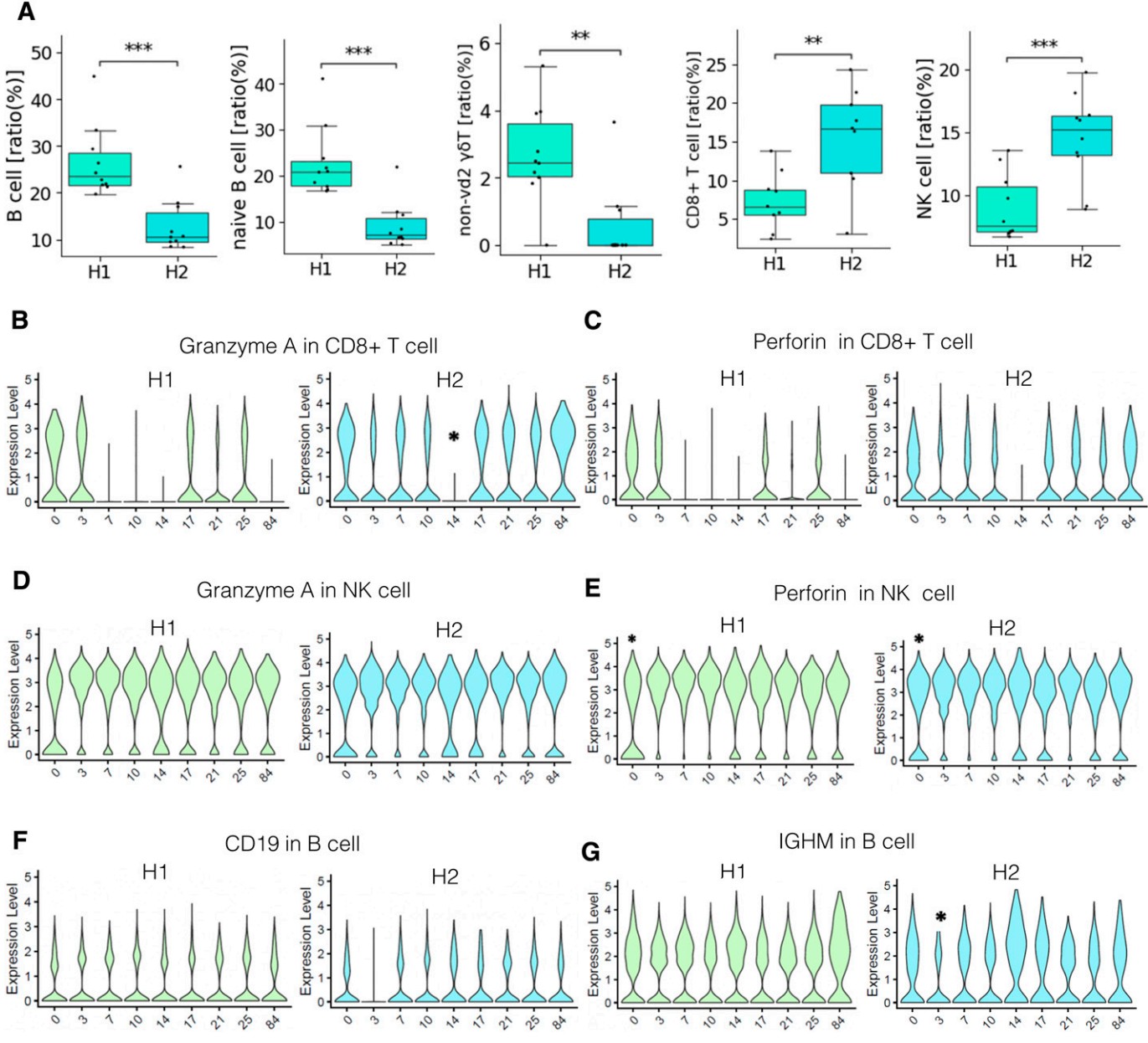

**Figure 2. Daily diversity of PBMC profiling in H1 and H2.**
**(A)** Ratio of each cell type in H1 and H2. Boxplots include eight time points of each person of B cell, naive B cell, non-vd2 gd T cell, CD8[+] T cell, and NK cell (left to right). *P*-value was calculated by a *t* test and shown as **: $1.00 \times 10^{-3} < P \le 1.00 \times 10^{-2}$, ***: $1.00 \times 10^{-4} < P \le 1.00 \times 10^{-3}$. **(B, C, D, E, F, G)** Expression level of representative genes in each cell type of H1 and H2; granzyme A expression of CD8[+] T cells (B), perforin expression of CD8[+] T cells (C), granzyme A expression of NK cells (D), perforin expression of NK cells (E), CD19 expression of B cells (F), and IGMH expression of B cells (G) in H1 and H2. *P < 0.05, compared with other eight time points. Adjusted *P*-values are used.

even within the same individual. Our result showed that higher proportions of B cells were detected in H1, a male donor, although a previous paper reported B-cell proportion is higher in females in middle-age (Klein & Flanagan, 2016) (Figs 2A and S3A). More specifically, naive B cells and non-vd2 γδ T cells were highly represented in H1 (Fig 2A). We further analyzed individual samples using the hemogram and blood chemistry test (Fig S3B). Our results suggested that H1 had a higher B cell ratio compared with other male donors (Fig S3A and B). It also showed that H1 have higly basophil ratio and seems to have IgE-mediated allergy (Fig S3C). On

the other hand, CD8[+] T cells and NK cells proportions were higher in H2 (Fig 2A and Table S3).Unlike usual T cells expressing α and β TCR chains, these non-vd2 γδ T cells do not necessarily require antigen representation via the MHC class I molecule for their activation (Weese et al, 2012), although their antigen recognition mechanism has not been fully characterized. These results suggested the possibility that the H1 immune landscape might be inclined to the humoral immune mechanism. On the other hand, the immune system in H2 can put a greater strain on the TCR-dependent response. Although some variations depended on the time points,

these differences were characteristic to the individuals with statistical significance (*P*-values are shown in panels and legend; Fig 2A).

We also characterized the activation states by measuring gene expressions across different time points using the representative cell types in H1 and H2 to understand how active the immune cells in healthy subjects. For representative active markers of CD8+ T cells and NK cells, we found that their expression levels were almost similar between H1 and H2, in spite that some daily changes were observed. Fig 2B–E exemplifies the case of perforin (Osińska et al, 2014) and granzyme A (Hayes et al, 1989; Shi et al, 1992). A similar observation was obtained for the activation state of B cells (Fig 2F and G). These results suggested that the difference between H1 and H2 under healthy conditions was rather represented by the number of the corresponding cells but not always the activation states of the individual cells.

## Immune-cell diversity in seven individuals of varying backgrounds

To further assess the diversity of cell compositions across individuals, we compared the cell type proportions of the other seven individuals (Fig 3A and B). The ninth (final) sample was taken as a representative for H1 and H2. As described above, H1 had a higher proportion of B cells than H2, and this trend even remained the most relevant among all samples (Fig 3A and B). Particularly, the plasmablast population was far higher than the average of the other samples (2.8% and 0.15% for H1 and H2, respectively). On the other hand, H3 and H4 showed even higher frequency of non-vd2 γδ T cells than H1, suggesting that this feature is not totally unique to H1. Furthermore, H2 had high representations of memory CD4+ T cells, effector CD8+ T cells and MAIT cells in H3, and vd2 γδ T cells in H4 (Fig 3B and Table S3). All individuals showed, in part similar, but a wide variety of unique features.

Among them, H7 showed a unique profile (Fig 3A). The cellular population and the gene expression profiles of individual cells suggested that NK cells are in the active state in this individual (Fig 3C and D). This individual is an older adult and has experienced malignant B-cell lymphoma (Fig 1A inset table). The cytotoxicity of NK cells has a high antitumor potential (Vivier et al, 2008). NK cells are often suppressed in blood cancer patients when the disease is in a malignant stage. However, as patients recover, the reactive population of NK cells increases, resulting in disease remission (De Kouchkovsky & Abdul-Hay, 2016). Although more than 5 yr have passed since the complete elimination of malignant B cells by successful R-CHOP chemotherapy, the remaining large proportion of NK cells in H7 may have expanded during therapy. A recent study reported that prolonged expansion of clonal NK cells occasionally occurs after recovery. In fact, a sustained expansion of NK cells may suggest clonal expansion because of response to any chronic stimulation (Adams et al, 2020) or acquisition of somatic mutations (Olson et al, 2021). Collectively, the results suggest that healthy individuals hold a prominent baseline immunological diversity.

Before further exploring the observed difference, we considered the validation using other methods to validate whether the observed diversity should correctly represent the diversity between individuals. For this purpose, cytometry of time-of-flight (CyTOF)

analysis was employed. This method uses the mass cytometer HeliosTM, using heavy metal isotope-tagged antibodies to detect PBMCs proteins at the single-cell resolution (Spitzer & Nolan, 2016). Four samples (H1, H2, H3, and H5) were subjected to the analysis (Fig 3E and F). Comparing the transcriptome and proteome datasets showed that the detected cellular compositions were roughly equivalent regardless of the analytical methods (Fig 3G). It is true that, the transcriptome data gave a larger inclination than the proteome data for some cell types, whereas the opposite was observed for other cell types (see Table S6 for details). Probably, these observations were because mRNA and protein levels are not strictly equal. Nevertheless, we found a high correlation between transcriptome and proteome data in almost all cases. Thus, the observed diversities of immune-cell profiles are validated from this viewpoint.

## Time-lapse changes of the immune landscapes in T-cell populations

We attempted to further characterize the diversity of immune-cell responses by considering the VDJ regions of TCR or their "clonotypes." Using the Chromium platform, the VDJ-seq libraries were constructed from the intermediate products of the library construction for the scRNA-seq (see Table S7 for the sequencing statistics). Because the cell barcodes were shared between the VDJ-seq and scRNA-seq libraries from the same sample, we could associate the observed VDJ information with the transcriptome information of its expressing T cell for each cell. Similarly, in the case of scRNA-seq, for H1 and H2, data were collected from nine points for H1 and H2 over a month (H1 day 0 to H1 day 84 and H2 day 0 to H2 day 84, Table S1).

For H1 and H2, even the 10 most frequent clonotypes claimed a small proportion of the overall annotated cell population (Fig 4A). Furthermore, the 10 most frequent clonotypes were unique to each individual, and no explicit overlap was observed (Fig 4B, see Table S8 for more details). Nevertheless, some features in the pattern of the compositions and their changes were commonly observed between H1 and H2 (Fig 4B). We examined and found that the clonotypes that were unique within the same individual over different time points ("sporadic" clonotypes) were mostly from naive T cells, probably representing a unique repertoire of unstimulated T cells in the individual (Fig 4C–F). On the other hand, as for the clonotypes detected from several time points, effector and memory CD8+ T cells were dominant (Fig 4C–F). Of note, from those "sustained" T-cell populations, MAIT cells accounted for a significant population (Fig 4D and F).

Interestingly, we could trace the time-lapse transition of their expressing T cells for some of those sustained clonotypes (Fig 4F). For example, for a particular clonotype, as shown in Fig 4G, its expressing T cells were effector CD8+ T cells. This proportion decreased within a week and memory CD8+ T cell appeared on day 21. This individual could have been infected with a pathogen in his/her self-presumed healthy state (Fig 4G, left). A similar situation was also confirmed in H2 (Fig 4G, right). Accordingly, the activation of T cell populations may constantly occur even in the "healthy" condition. We further searched from H1 and H2 and identified a total of 85 and 209 clonotypes which increased and decreased during this time-frame. Immune cells may undergo constant changes,

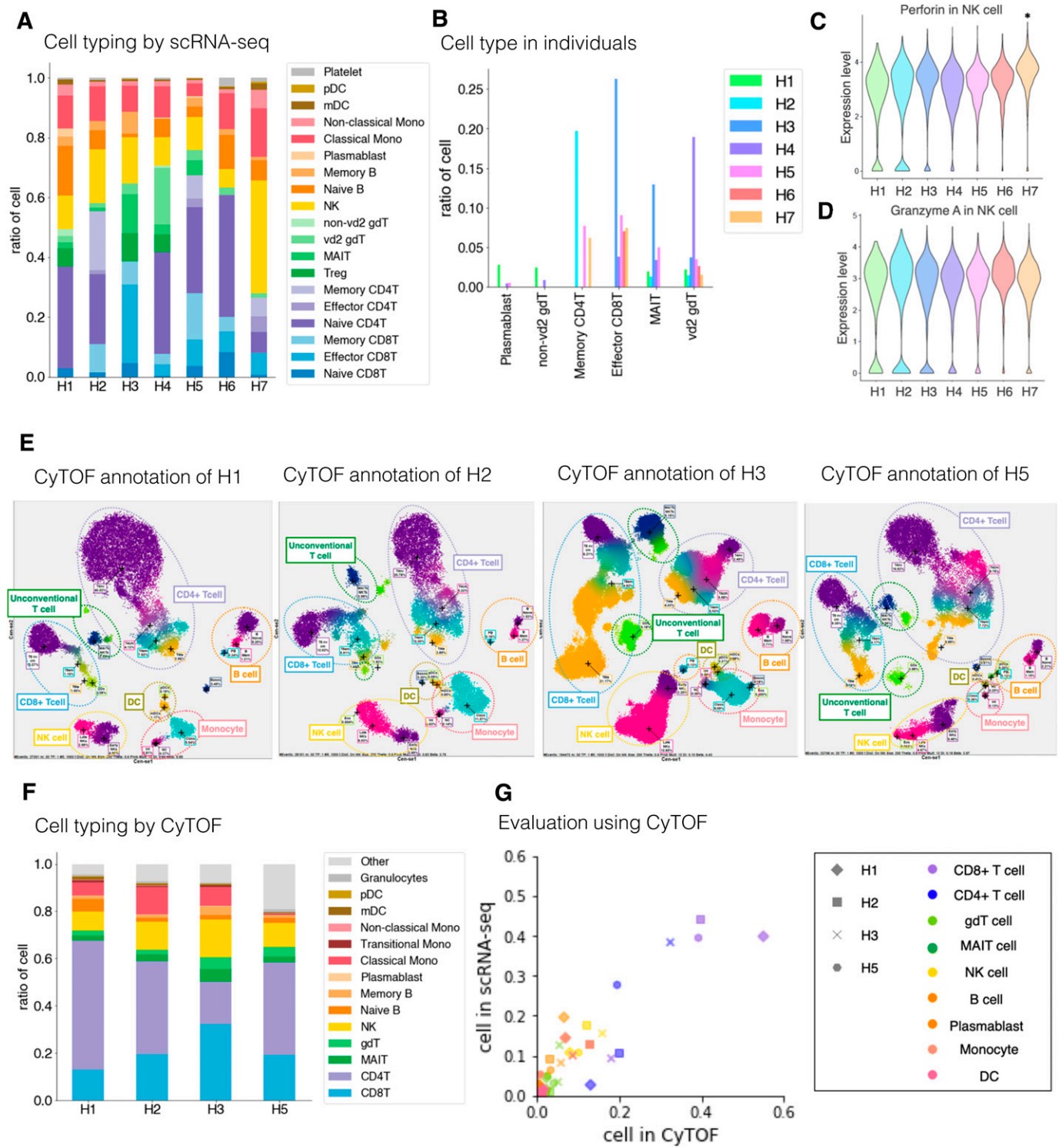

**Figure 3. Diversity of PBMC profiling in seven individuals.**
**(A)** Structure of PBMC in seven individuals. Cells are annotated based on the gene expressions in the dataset analyzed by scRNA-seq. The *x-axis* shows the individuals, and the y-axis shows the ratio of each cell component. Color legends are shown in the margin. **(B)** Individual variance of cell types. The *x-axis* shows the focusing cell type, and the *y-axis* shows the ratio of cells in individuals. **(C, D)** Gene expression level in seven individuals. Gene and cell type of interest are shown at the top of the graph. **(C)** Expression of perforin in H7 NK cells is statistically significant compared with other individuals (C). **(D)** Expression of granzyme A in H7 NK cells is not significant compared with other individuals (D). *$P < 0.05$. no mark: n.s. Adjusted *P*-values are used. **(E)** Cell typing using CyTOF of H1, H2, and H5 (from left to right). **(F)** Structure of PBMC based on CyTOF. The *x-axis* shows the individuals, and the y-axis shows the ratio of each cell type. Color annotations are shown in the margin. **(G)** Evaluation analysis using CyTOF. Scatterplot showing the correlation of the ratio of cells annotated by CyTOF (*x-axis*) and by scRNA-seq (*y-axis*). Markers are shaped depending on each individual and colored by cell types shown in the margin. For H1 and H2 used in Fig 3, we selected final time point as representative samples.

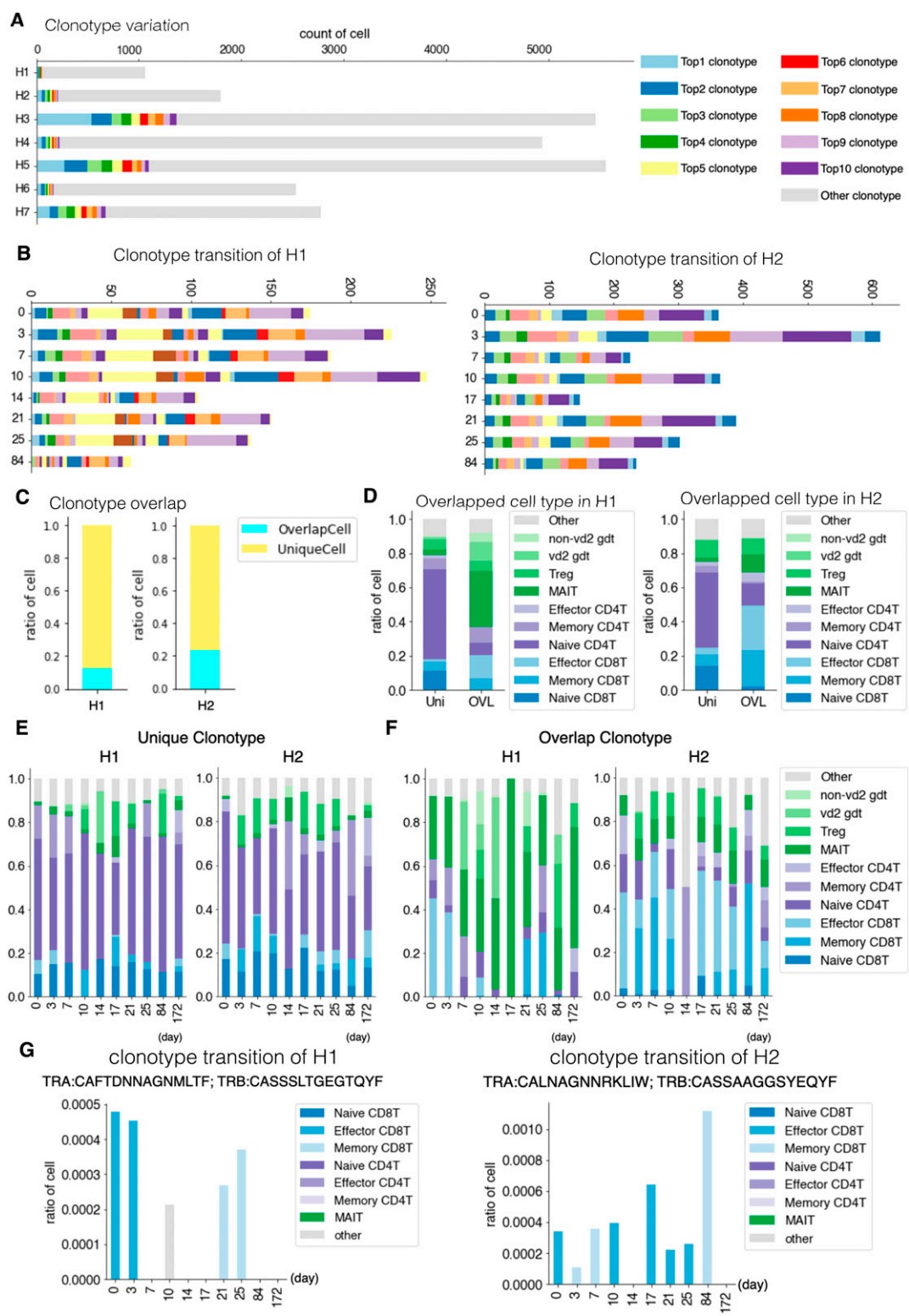

**Figure 4. TCR individual and daily variation.**
**(A)** Clonotype divergence between seven individuals. The *x-axis* shows the number of T cells, and the *y-axis* shows the individual. Cell with top 1 to top 10 clonotypes are plotted as specific colors shown in margins. Top clonotypes shown are not common between individuals. **(B)** Clonotype divergence of eight time points of H1 (left) and H2 (right). Barplot shows clonotypes ranked as top 1 to top 10 at any point in time. Note that color in H1 and H2 is not common. **(C)** Barplot showing the ratio of T cells with unique TCR (light blue) and overlapped TCR (yellow) in H1 (left) and H2 (right). **(D)** Barplot showing the ratio of the detailed T-cell type in unique (Uni) and overlapped (OVL) of H1 average (left) and H2 average (right). **(E, F)**, Barplot showing ratio of detailed T-cell type in unique (E) and overlapped (F) of H1 (left) and H2 (right). **(G)** Barplot showing the ratio of T cells at each time point with specific clonotype TCR of H1 (left) and H2 (right). Information about the TCR is shown at the top of each graph.

responding to environmental epitopes, and such responses may have shaped the unique immune landscape of the individual over long years.

## Diversity of TCRs and searching for their possible epitopes

We conducted a similar TCR analysis for the other individuals. Despite the reduced data points for other samples, similar trends were also observed for H4 and H6, although their exact VDJ sequences were, again, unique to the individuals (Fig 4A). Of note, in H3, the most and the second-most frequent clonotypes claimed were clonotype 1 (13.4%) and clonotype 2 (4.8%), respectively (Fig 4A). We carefully ruled out the possibility that these were derived from PCR and other artifacts by manually inspecting the correct assignment of cell barcode and unique molecular index. Particularly for this clonotype, we dissolved their TCR states by using the scRNA-seq information. We found that clonotype 1 and 2 were mostly for effector CD8$^+$ T cells, suggesting that some asymptomatic infection events are ongoing (Fig S4A).

As for H3, this individual is originally from a suburb in Indonesia (Fig 1A inset table). Considering the country of his/her upbringing, we postulated that this individual may have frequently experienced the infections of CMV, EBV, and other common pathogens. We conducted the intracellular cytokine staining assay (ICS assay) for H1, H2, and H3 samples (Fig S4B). We found that H3 showed the highest response for the CMV stimulation. Similar responses were found for EBV (Fig S4B). In H3, the immune system generally may remain alerted, which could be a common trait for individuals originally from developing countries. Supportingly, when we attempted to infer potential epitopes for the clonotypes detected in H1–H3, using the deduced amino acid sequences of the CDR3 region, which is the docking platform for the epitopes; for the bioinformatics prediction pipeline and epitope databases TCRex (Gielis et al, 2019) and VDJdb (Shugay et al, 2018), we found that CMV and EBV appeared to be potential candidates for the TCRs more frequently in H3 (Tables S9).

## Diversity of BCRs

We conducted a similar analysis for BCRs (sequence statistics are shown in Table S7). Even to a lesser extent than the TCRs, the BCR clonotypes did not overlap within the same individual over time, particularly for the usage of the immunoglobulin heavy chain (99% were uniquely observed; Fig 5A). The major unique clonotypes were mostly for IgH-M in H1, suggesting that there is constant activation of B cells for possible novel antigens in this individual (Fig 5B). On the other hand, IgH-G was more relevant within a minor population of overlapping clonotypes at different time points and possibly represented the sustained activation of the corresponding clonotypes. A similar trend was observed for H2 (Fig 5B) but to a lesser extent than H1. We further examined the overall entropy of the BCRs. The distribution of the Shannon index showed that H1 had the higher entropy than H2 and the other individuals (Fig 5C), although daily changes were observed in this aspect (Fig 5D). As well as TCRs, diversity of BCR in naive B cells are considered to be higher than that of memory B cells, which have already experienced clonal expansion by specific antigen stimulation. The higher entropy of BCRs in H1 may reflect higher frequency of naive B cells. When we

analyzed the frequency of the variant regions, we found that the H1 showed a focused use of particular variant types, although their precise clonotypes were diverse (Fig 5E). These results showed that BCR profiles also vary between individuals. They also collectively indicated, again, that B cell–mediated immune responses are prominent in H1.

## Influence of the vaccination on the immune-cell profiles

We considered vaccination as an ideal usual life event to further characterize the personal immune landscapes and their changes. During vaccination, the exact antigen is defined, and the exposure time is known. We first collected PBMC samples from H1 and H2 before and after influenza virus vaccination (antigen was for the 2020, see the Materials and Methods section). Relevant antibodies titers were confirmed by the antibody quantification method (Table S10). Samples were collected on day-1, 1, 3, 7, and 28 post vaccination (dpv, day post vaccination) and subjected to similar scRNA-seq and VDJ-seq analyses for TCRs and BCRs (Fig 6A, see Tables S2 and S7 for the sequencing statistics and cell annotation details for Fig S5A–D).

Again, diverse immune-cell profiles between different individuals and time points were found for this time-lapse dataset. Significant differences were observed in response to the vaccination between H1 and H2 (Fig 6B). The distinct responses appeared, representing their original immune landscapes (see below). Nevertheless, several common features were observed, generally consistent with previous knowledge describing general features of immune-cell responses. For example, expansion of monocytes, primarily CD14$^+$ classical monocytes, were detected as the primary responder of the stimulation immediately after vaccination. This induction was followed by a temporary reduction of naive B cells in PBMC (Fig 6B). CD4$^+$ T cells and $\gamma\delta$ T cells were also temporarily reduced for T-cell populations, whereas CD8$^+$ T cells, NK cells, and DCs cells retained their original population sizes. For these profiles, we also conducted the validation analysis using CyTOF and confirmed the robust representation of the observed results (Fig S6A–C). Of note, these initial responses were recovered to their original levels by day 28 post vaccination, when the immune responses are estimated to be complete and memory cells established (Fig 6B).

Although the above responses were applied to H1 and H2 in general, several unique features appeared to depend on the individual. For example, in H1, the population of MAIT cells was particularly reduced from PBMC at the initial response from 2.7% to 0.2% in scRNA-seq (2.6–0.2% in CyTOF) (Fig S6D, top right). In addition, the response of CD4$^+$ T cells was more pronounced than that of H2 (Fig S6D, bottom center), whereas the population of CD8$^+$ T cells was larger at all times (Fig S6D, bottom right), perhaps recapturing the dominant humoral responses in H1 (Fig 6B).

We inspected the changes of TCR clonotypes in response to vaccination (Figs 6C and D and S7). The TCR repertoire and clonotypes detected before vaccination were removed to focus on the specific response to the vaccination. A total of 10 VDJ sequence datasets were used for the subtraction for each individual (see Table S7 for sequence the statistics).

For the remaining clonotypes obtained, we attempted to identify the clonotypes showing dynamic changes in response to the

**Figure 5. Individual variation and daily variation of the B-cell receptor (BCR).**
**(A)** BCR clonotype divergence in H1 (left) and H2 (right). Barplot showing the ratio of overlapped and unique clonotype of IgH, IgL, and IgK. Color legend is shown in the margin. **(B)** Ratio of each clonotype of unique (left) and overlap (right) in H1 and H2. Color legend is shown in the margin. **(C)** Shannon index score variation of BCR in H1–H7. Color legend is shown in the margin. For H1 and H2, we used the average score of day 0 to day 84. **(D)** Daily variation of the Shannon index score of BCR daily variation in H1 (left) and H2 (right). The *x-axis* shows the time point, and the *y-axis* shows Shannon's score. **(E)** Variation of V gene in BCR from day 0 to day 84 (top to bottom) in H1 (left) and H2 (right).

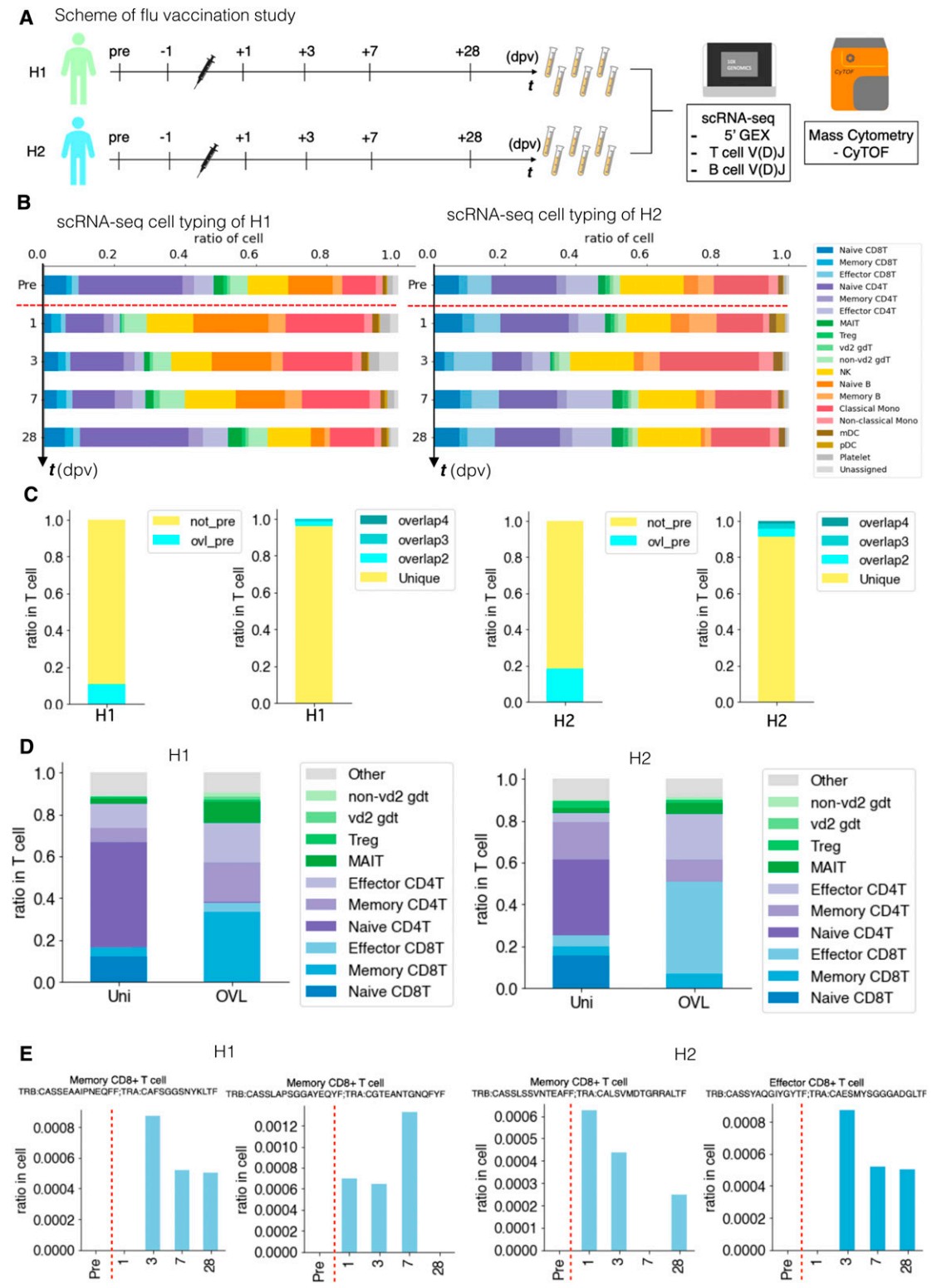

**Figure 6. Perturbation of the immune-cell gene expression profiles depending on influenza vaccination.**
**(A)** Influenza virus vaccination study scheme. We collected blood samples from H1 and H2 1 d before vaccination (−1 dpv, day postvaccination), 1, 3, 7, and 28 dpv. We analyzed PBMCs transcriptome, V(D)J of B-cell receptor and TCR at the single-cell level using 10x Genomics and mass cytometry at the single-cell level using Fluidigm CyTOF. **(B)** Barplot showing the transition of cell components before and after influenza vaccination of H1 (left) and H2 (right) analyzed by scRNA-seq. **(C)** Barplot showing the ratio of T cells with unique TCR and overlapped TCR in H1 (left) and H2 (right). Clonotype exit in pre-vaccination is annotated as "ovl pre" and in only after vaccination annotated as "not pre." Clonotype of "not pre" further grouped into unique, only exit at one time point, and overlap (ovl). The number after ovl shows number of

vaccination (Fig 6E). Again, we observed that some clonotypes sporadically appeared in a particular time point; others were persistent. We compared the T-cell populations between those sporadic and persistent populations. We found that the CD4⁺- and CD8⁺-naive T cells were characteristic in the sporadic population, suggesting that these could be the clonotypes that were not removed by the subtraction. On the other hand, CD4⁺ and CD8⁺ memory T cells as well as MAIT cells were more relevant in the persistent population, suggesting that the clonotypes first induced by vaccination were enriched in this population. Some examples are shown for the clonotypes which showed dynamic changes (Fig 6E) as the candidate T cells first induced in response to vaccination. At least, a total of 16 and 53 such clonotypes were detected, in H1 and H2, respectively.

As for the BCR repertoire, we did not find any relevant overlapping clonotypes as shown in the above analysis (Fig S8A and B). When we examined their complexity (Fig S6C), we found that the entropy score increased on day 1 as an immediate response of BCRs. This induction recovered to the original level gradually by day 28. To different extent, this trend was commonly observed for both H1 and H2 (Fig S6C). The variant analysis also showed that the particular variants were induced on day 1 (Fig S6D). These results indicated that the B-cell system was also responding to vaccination, again to varying degrees in different individuals.

### Responses of TCR clonotypes in response to the SARS-CoV-2 vaccinations

We conducted a similar analysis for vaccination against SARS-CoV-2 (Fig 7A, see Tables S2 and S7 for the statistics, and Fig S9A–F for detailed cell annotation). This time, the mRNA vaccine of BNT162b2 produced by Pfizer-BioNTech was considered. PBMC samples from H1, H6, and H7 individuals were used for analysis (Fig 7A). Relevant increases in the antibody titers were measured by standard antibody quantification (Fig 7B and C and Table S11).

Similar features with influenza vaccination were observed here. These responses were also generally consistent with the results of a recently published paper (Ewer et al, 2021). These features include the immediate induction of monocytes and eventual restoration of the immune-cell states (Figs 7D and S10 for the CyTOF datasets). Those changes were more significant than the case of the Influenza virus vaccination, possibly reflecting more intensive nature of the SARS-CoV-2 vaccine. Again, we examined and found that the strength and the timing of such responses depend on the original immune landscapes (see below for an exceptional case of H7). Notably, the initial induction of monocytes was generally higher with SARS-CoV-2, perhaps, consistent with the fact that inflammatory side effects of this vaccine, such as fever and inflammation, in this individual was stronger than the influenza vaccine. It should also be noted that, in this case, CD16⁺ non-classical monocytes were also induced, followed by the induction of CD14⁺ classical monocytes, indicating an enhanced immune response of this vaccination.

In the second vaccination of H1 and H6 (Fig S9G and H), the same response as mentioned above was observed.

We particularly attempted to inspect the changes of clonotypes after SARS-CoV-2 vaccination (sequencing statistics shown in Table S7). The pre-vaccination TCR clonotypes were collectively removed. To particularly focus on the responses specific to the SARS-CoV-2 vaccine, all the VDJ sequences observed for influenza vaccination analysis were also subtracted (Figs 7E and S11A and B). Consistent with influenza vaccination, the most of the clonotypes were only detected at a single time point, reflecting the complexity of the TCR population. Nevertheless, a total of 20 and 62 clonotypes were identified from more than three-time points, in H1 and H6 datasets. Similar to influenza vaccination, characteristic subpopulations of the sporadic and persistent populations were also observed with SARS-CoV-2 vaccination (Figs 7E and S11A and B).

To further characterize these sporadic and persistent populations, we examined the gene expression level of CD69 as a marker for activation of T cells. We found that the T cells of the persistent group showed higher CD69 levels, suggesting that those cells were in an active state (Fig S11C and D). We further traced their time-lapse changes and identified several clonotypes that were induced at some time points. Most of those T cells were induced at the first vaccination and then enhanced further by the second vaccination (Figs 7F and S11E). These TCRs may represent the cells that were specifically induced by this vaccination.

Interestingly, H7 showed an overall unique character. As described above, this individual originally had a higher percentage of NK cells. When we examined the changes in the immune-cell profiles in H7 (Figs 7D and S9H), the changes in the immune-cell profiles were less relevant than H1 and H6. This observation may reflect the advanced age of this individual or a generally high level of NK cell–centered immune-cell activity in its pristine state, possibly based on that individual's medical history (Fig 1A inset table). Although this individual eventually acquired a sufficient antibody level, the level obtained was, to some extent, lower than H1 and H6 (Fig 7B and C). Consistently, the changes of the CD69 levels were less significant, which in turn suggest that the vaccine responses depend on the original immune state of the individuals (Fig S11C and D).

## Discussion

In this study, we attempted to describe the diversity of the immune-cell profiles in PBMC among healthy individuals. We revealed that the gene cellular components and gene expression profiles are diverse even in healthy individuals, possibly reflecting the personal history of previous immune responses. The unique point of this study is that we employed intensive scRNA-seq transcriptome and TCR-VDJ sequencing analyses validated by single-cell mass spectrometry analysis. This approach could track a single clonotype across the sampling time points in association with its expressing

appearance. **(D)** Barplot showing the ratio of the detailed T-cell type in unique and overlapped of H1 average (left) and H2 average (right). **(E)** Barplot showing the ratio of T cells with specific clonotype TCR of H1 (left) and H2 (right) in each time point. The *x-axis* shows day postvaccination, and the *y-axis* shows the ratio of specific T cells. Information about the TCR clonotype is shown at the top of each graph.

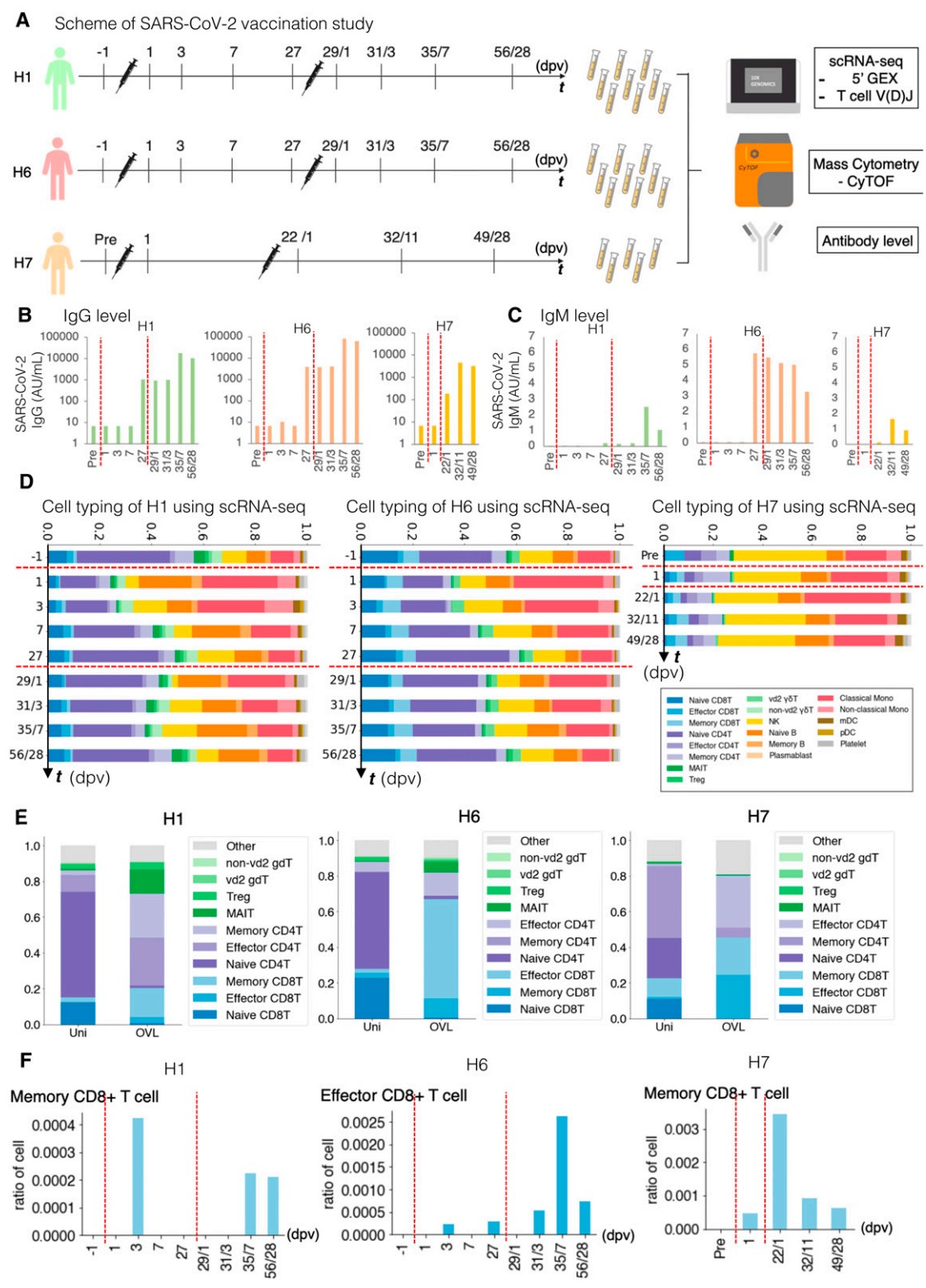

**Figure 7. Perturbation of the immune-cell gene expression profiles depending on SARS-CoV-2 vaccination.**
**(A)** SARS-CoV-2 vaccination study scheme. Blood samples of H1, H6, and H7 were collected before the vaccination (1 d before vaccination, −1 dpv), post–first vaccination, and post–second vaccination. We analyzed PBMCs transcriptome, V(D)J of TCR at the single-cell level using 10× Genomics and proteomics at the single-cell level using Fluidigm CyTOF and anti-SARS-CoV-2 virus antibody level. **(B, C)** Transition of anti-SARS-CoV-2 antibody levels. Barplot shows the antibody level at each time point. **(B, C)** The *x-axis* shows the day after the first vaccination/second vaccination, and the *y-axis* shows the level of IgG (AU/ml) (B) and IgM (AU/ml) (C) in H1 (green, left), H2 (coral, middle), and H7 (orange, right). **(D)**, Barplot shows the transition of cell components before and after SARS-CoV-2 vaccination based on scRNA-seq datasets of H1 (left), H6 (middle), and H7 (right). **(E)**, Barplot shows T-cell component transitions of H1 (left), H6 (middle), and H7 (right). **(F)** Barplot showing the ratio of CD8⁺ T cells with specific TCR in SARS-CoV-2 vaccination in H1 (left), H6 (middle), and H7 (right). The *x-axis* shows the day postvaccination, and the *y-axis* shows the ratio of the specific T cells.

T-cell state. To the best of our knowledge, although there may be some previous studies which have analyzed immune-cell profiles of healthy individuals, those studies used the Western population. In this study, we considered the Asian populations, which are supposed to show distinct immune responses to various pathogens, including SARS-CoV-2. By collecting and reanalyzing the previous data for the Western population and comparing it with the data of the present study, the immunological difference in health status will also be unveiled. Such insight is particularly of interest, considering that in the early stages of the COVID-19 pandemic, allegations were made that ethnicity may be responsible for the variability in susceptibility and morbidity worldwide (Barash et al, 2020; Bunyavanich et al, 2020; El-Khatib et al, 2020; Hou et al, 2020; Sze et al, 2020). Also, the difference in immune responses is also associated with the effectiveness of the vaccination. There are some papers describing the cause of such ethnic differences as the pre-existing discrepancies in health equity, such as access to healthcare and social determinants of health, and likely not genetics (Lee et al, 2020; Shelton et al, 2021). The ethnicity effects on immune response should also consider their medical records and the current environments.

The obvious drawbacks of the present study include the general lack of in-depth biological validations. In particular, the results of the epitope identification were not validated for various pathogens. More generally, even after long discussions, the extent to which PBMC should represent the immune states of the individual remains debatable. Also, the small sample size and especially the short sampling period also limit the comparability of the results to previous in vitro laboratory studies. Particularly for the immune response to the vaccination, careful analyses are needed to elucidate what molecular events are occurring there in more detail. However, it is generally technically difficult to validate the unique events taking place in individuals over time of their personal histories.

Nevertheless, it is significant that we could identify the individual heterogeneity of healthy immunity. The main aim of this study was to generate a base for such future studies to address the issues named above. In particular, the interindividual heterogeneity was more pronounced than the intraindividual temporal variance. Therefore, future studies investigating immune system fluctuations in disease should account for the baseline diversity among healthy individuals, as demonstrated in this study.

No less important, we consider that the present study results have indicated the importance of data collection for particular individuals. The immune-cell profiles should be so diverse that in the event of a disease, the healthy state information should be directly subtracted, and the status of the immune cell analyzed. It is important to know the state of immune cells for infectious diseases and various types of other diseases, such as cancers. Recently, many anticancer drugs are designed to control proper or enhanced actions of the immune cells (June et al, 2018). Importantly, once the disease develops, the profile of the healthy states would be lost; thus, such information should be collected beforehand. This direction should be followed by the need for "personal immunological records." The records may include not only the data resource but also the banked biomaterial samples.

Personal health or medical histories, which differ depending on the immune responses experienced throughout their lives, should have collectively shaped their current immune landscapes. Such a landscape is the base to determine his or her unique immune condition in the daily life or to predict or control his or her response to various diseases. Therefore, the "personal immune landscape" may become the data resource which should be prepared ideally for each individual. The present study should have paved the first step toward the new era of "personalized genomics" research and its social applications.

# Materials and Methods

### Ethics approval and consent to participate

The human materials were collected and analyzed after the procedure approved by the ethical committee of the University of Tokyo as examination number: 20-351. All human subjects provided written informed consent.

### Library preparation and sequencing of single-cell analysis

PBMCs samples were collected from seven healthy donors (list of samples used in the current study is summarized in Table S1). Two participants, H1 and H2, had their PBMCs collected nine times over a month (Fig 1A). A single sample was collected for the other participants. Except for the first sample from H1 and H2, samples were frozen and thawed before processing. Each sample was processed with the Chromium Next GEM Single Cell 5′ Library Construction and Gel Bead Kit following the manufacturer's user guide (10× Genomics; PN-1000165, PN-1000020, PN-1000120). Single-cell suspension was prepared to target 10,000 cells/sample and loaded with barcoded gel beads and partitioning oil to generate GEM. After cDNA amplification, the B-cell and T-cell V(D)J were enriched using the human B-cell and T-cell enrichment kit for TCR and BCR library construction (10× Genomics; PN-1000005, PN-1000006). The prepared 5′ GEX, TCR, and BCR libraries were then sequenced using the Illumina NovaSeq sequencer according to the manufacture's instruction. The sequence was conducted 28/91 bp PE both for 5′GEX and TCR/BCR libraries.

### Single-cell 5′ GEX analysis using Seurat

The 5′ GEX dataset was initially processed with Cell Ranger (v.3.1.0 for the daily variation study and version v5.1.0 for influenza and SARS-CoV-2 vaccination study) and underwent quality control and clustering by the R package Seurat (version 3.2 for daily variation study and version 4.1 for influenza and SARS-CoV-2 vaccination study) (Stuart et al, 2019). Downstream analysis is conducted based on Seurat vignettes. Briefly, for quality control, nFeature_RNA, number of molecules detected within a cell, and percent.mt, the percentage of reads that map to the mitochondria gene, are used. We filter cells that have nFeature_RNA over 4,000 or less than 200 and >8% mitochondrial counts with the command: subset (SeuratOBJ, subset = nFeature_RNA > 200 & nFeature_RNA < 4,000 & percent.mt < 8). For clustering of multiple samples merged datasets, we performed "integrated" analysis (for clustering, we used pcadim = c(1:30), resol = 0.5). For expression level analysis, we switched back

to the original data using "RNA" slot. *P*-value shown in the violin plot is based on the expression level after scaling.

### Cell type annotation in sequencing dataset

We employed SingleR (Aran et al, 2019) and Azimuth (Hao et al, 2021) to reference annotation. This study used a publicly available bulk RNA-seq dataset of sorted immune cells as a reference for SingleR imputation (Monaco et al, 2019). After primarily annotation, we confirmed the correspondence of each cluster and cell type annotation with differentially expressed gene set and the canonical markers retrieved from the literature (Tian et al, 2019; Martos et al, 2020). For T-cell detailed annotation, we reclustered cells. The expression level of the representative genes used for the cell type annotation is shown in the Fig S1 (daily and individual variation studies), Fig S2 (influenza vaccination study), and Fig S3 (SARS-CoV-2 vaccination study). T cells were identified based on *CD3D* and were determined as CD8$^+$ or CD4$^+$ depending on *CD8A and CD8B* or *CD4* expression, respectively (Hu et al, 2020). CD8$^+$ T cells were further classified into effector (*GZMK, GZMH, PRF1,* and *CCL5*), memory (*CD29*), and naive (*CCR7*). CD4$^+$ T cells were similarly classified into naïve (*IL7R, CCR7*), memory (*IL7R* and *S100A4*), and Treg (*FOXP3* and *IL2RA*) (Szabo et al, 2019). Other T cells included MAIT cells (*SLC4A10* and *TRAV1-2*) and γδ T cells (*TRGV9* and *TRDV2*). B cells were identified (*CD79* and *CD79B*) and were labeled naïve based on *CD27* (Hu et al, 2020). Plasma B cells were based on SDC1 and *MZB1$^+$/ XBP1$^+$* (Zhao et al, 2020). Monocytes were grouped into either classical monocytes (*CD14* and *LYZ*) or nonclassical monocytes (*FCGR3A* and *MS4A7*). DCs identified based (*IL3RA, CLEC4C, CST3,* and *NRP1*) on and were typed as myeloid DCs (*FCER1A* and *CD1C*) and plasmacytoid DCs (*FCER1A* and *LILRA4*) (Hu et al, 2020). We further identified NK cells (*KLRB1, KLRC1, KLRD1, KRRF1, GNLY, NKG7,* and *CD56*) and platelets (*PPBP*).

### T-cell receptor and B-cell receptor analysis

For scVDJ-seq of BCR and TCR, data were processed using Cell Ranger (v.3.1.0 for the daily variation study and version v5.1.0 for the influenza and SARS-CoV-2 vaccination study). After Cell Ranger processing of T-cell datasets, a list of the CDR3 sequence and TCR gene was generated. To acquire detailed T-cell annotation, TCR datasets are combined with 5' gene expression datasets. The antigen prediction for each clonotype was conducted by VDJdb. For BCR analysis, data were analyzed by scRepertoire (Borcherding et al, 2020).

### Analysis using mass cytometry

For the Helios Mass Cytometer (Cat. no. 107001; Fluidigm, sample list and their biological descriptions are shown in Table S1), we used a part of PBMC samples used for single-cell analysis. To characterize detailed cell type of PBMCs, we applied Maxpar DirectTM Immune Profiling AssayTM (Cat. no. 201235; Fluidigm), commercial preset kit including 30 markers and processed following vendors' guide (Quick Reference guide JPN_PN 400288 B1_001). Briefly, counted PBMCs are washed with cell staining buffer and processed with FcX (Cat. no. 422301; BioLegend). Cells are stained with 30 antibodies included in the kit. Description of the used detectors, including, the

list of marker antibodies, channels and clone information are shown in Table S12. After cell staining, cells are fixed with 1.6% formaldehyde (Cat. no. 28906; Thermo Fisher Scientific). Stained cells are analyzed with Helios, a CyTOF system. Datasets deposited to the National Bioscience Database Center and our private database are shown in the Data access section. Acquired datasets were analyzed with Maxpar Pathsetter (Cat. no. 401018; Fluidigm). After Maxpar Pathsetter processing, Maxpar Pathsetter reports showing count of cell and Cen-se' map were generated.

### Antibody titration

In the influenza vaccination study, we measured the antibody titer for type A H1, type A H3, type B Yamagata, and type B Victoria flu using HI method (Table S10). In the SARS-CoV-2 vaccination study, we measured the antibody titer anti-SARS-CoV-2 S IgG and anti-SARS-CoV-2 S IgM (Table S11).

### ICS assay using flow cytometry

In the ICS assay, we stimulated PBMC of H1, H2, and H3 with CMV (pp65 and IE1) and EBV (EBNA1, LMP1, and BZLF1) and incubated for 6 h in 37°C. For staining, we used FITC (CD4, Cat. no. 317408; BioLegend, RRID: AB_571951), PE (CD107a, Cat. no. 328608; BioLegend, RRID: AB_1186040), PerCP (CD8a, Cat. no. 300922; BioLegend, RRID: AB_1575072), PE-Cy7 (IL2, Cat. no. 500326; BioLegend, RRID: AB_2125593), APC (TNFa, Cat. no. 502912; BioLegend, RRID:AB_315264), APC-Cy7 (IFNg, Cat. no. 502530; BioLegend, RRID: AB_10663412), Pacific Blue (CD3, Cat. no. 300330; BioLegend, RRID:AB_10551436), and LIVE/DEAD Aqua-Amcyan Antigen for staining cells. We employed BD FACSCanto (BD), and the sort logic was set by gating lymphocytes by forward scatter and side scatter and then gating on CD3$^+$ CD4$^+$ cells and CD3$^+$ CD8$^+$ cells. The dataset was analyzed by FlowJo software version.10.

### Clinical blood test

Using blood samples of healthy donors, we conducted a clinical blood test. For H1 and H2 longitudinal serum samples, we measured a nonspecific IgE level (Fig S3C). Samples are collected at the same time point as daily variance cohort, day 3, day 7, day 10, day 14, day 17, day 21, day 25, and day 84 (healthy, not vaccinated period). Those samples are analyzed by the CLEIA method (JLAC10: 5A090-0000-023-052, SRL Inc). For whole blood samples of H1, H2, H5, and H6, we conducted hemogram analysis (Fig S3B). Samples are collected to a tube including EDTA-2K (Cat. no. VP-DK-052K05; Terumo). The number of cells are counted by automated blood cell counter and typed into neutrophil (standard: 40.0–74.0%), eosinophil (standard: 0.0–6.0%), basophil (standard: 0.0–2.0%), monocytes (standard: 0.0–8.0%), and lymphocytes (standard: 18.0–59.0%) (JLAC10: 2A160-0000-019-309, SRL Inc).

## Data Availability

The raw data have been deposited to the National Bioscience Database Center as study number: https://ddbj.nig.ac.jp/resource/jga-study/JGAS000321 (https://ddbj.nig.ac.jp/resource/jga-study/JGAS000321).

Metadata of deposit datasets are included in Table S13 (scRNA-seq), S14 (scVDJ) and S15 (mass cytometry) following FAIR principles. Details of deposited data are shown below. JGA accession number of the study: https://ddbj.nig.ac.jp/resource/jga-study/JGAS000321, JGA accession number for dataset: https://ddbj.nig.ac.jp/resource/jga-dataset/JGAD000432, study title: Comprehensive analysis of interaction between human gene expression and environmental metagenomes (https://ddbj.nig.ac.jp/resource/jga-study/JGAS000321), dataset title: WGS, 10× scRNA, 10× TCR, 10× BCR and CyTOF data of blood samples (https://ddbj.nig.ac.jp/resource/jga-dataset/JGAD000432), number of samples: 153, number of data files: 459 (fastq:459), number of analysis files: 30 (tab:30), file size: 2.2 TB (2,157,143,498,962 bytes), and usage restriction policy: NBDC Policy (https://ddbj.nig.ac.jp/resource/jga-policy/JGAP000001). In addition to the NBDC database, we also put processed dataset on our private database (https://kero.hgc.jp/longread_viewer/single_cell/open/data/Expression/). The present study did not develop any new software. All codes used in the present study can be available upon request to the lead contact, Yutaka Suzuki (ysuzuki@k.u-tokyo.ac.jp).

## Supplementary Information

## Acknowledgments

We thank all the anonymous donors who participated in this study. We appreciate Shintaro Yanagimoto for supporting immunological studies. We thank K Imamura, K Abe, Y Ishikawa, M Konbu, E Kobayashi, E Ishikawa, and S Minamiguchi, Y Kuze for their technical assistance. This work was supported by JST Moonshot R&D–MILLENNIA Program Grant Number JPMJMS2025.

### Author Contributions

Y Kashima: investigation, visualization, and writing—review and editing.
K Kaneko: visualization and writing—review and editing.
P Reteng: investigation and writing—original draft, review, and editing.
N Yoshitake: investigation and writing—original draft.
LR Runtuwene: resources and writing—review and editing.
S Nagasawa: data curation and writing—review and editing.
M Onishi: data curation and writing—review and editing.
M Seki: writing—original draft, review, and editing.
A Suzuki: writing—original draft, review, and editing.
S Sugano: supervision and writing—review and editing.
M Sakata-Yanagimoto: writing—original draft, review, and editing.
Y Imai: writing—original draft, review, and editing.
K Nakayama-Hosoya: investigation and writing—review and editing.
A Kawana-Tachikawa: visualization and writing—review and editing.
T Mizutani: writing—original draft, review, and editing.
Y Suzuki: conceptualization, supervision, funding acquisition, project administration, and writing—original draft, review, and editing.

## Conflict of Interest Statement

The authors declare that they have no conflict of interest.

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
