## [Reviewer comments · Life Science Alliance]

Life Science Alliance

Intensive Single Cell Analysis Reveals Immune Cell Diversity among Healthy Individuals

Yutaka Suzuki, Yukie Kashima, Keiya Kaneko, Patrick Reteng, Nina Yoshitake, Lucky Runtuwene, Sato Nagasawa, Masaya Onishi, Masahide Seki, Ayako Suzuki, Sumio Sugano, Mamiko Sakata-Yanagimoto, Yumiko Imai, Kaori Nakayama-Hosoya, Ai Kawana-Tachikawa, and Taketoshi Mizutani

DOI: <https://doi.org/10.26508/lsa.202201398>

Corresponding author(s): Yutaka Suzuki, University of Tokyo

Review Timeline:	Submission Date:	2022-02-03
	Editorial Decision:	2022-02-03
	Revision Received:	2022-03-17
	Editorial Decision:	2022-03-17
	Revision Received:	2022-03-22
	Accepted:	2022-03-22

Transaction Report:

Please note that the manuscript was previously reviewed at another journal and the reports were taken into account in the decision-making process at Life Science Alliance. Since the original reviews are not subject to Life Science Alliance's transparent review process policy, the reports and author response cannot be published.

February 3, 2022

Re: Life Science Alliance manuscript #LSA-2022-01398

Yutaka Suzuki

Dear Dr. Suzuki,

Thank you for submitting your manuscript entitled "Intensive Single Cell Analysis Reveals Immune Cell Diversity among Healthy Individuals" to Life Science Alliance. We invite you to submit a revised manuscript addressing the Reviewer comments, excluding Reviewer 3's request for deeper analysis to ask specific addressable biological questions.

Thank you for this interesting contribution to Life Science Alliance. We are looking forward to receiving your revised manuscript.

Sincerely,

B. MANUSCRIPT ORGANIZATION AND FORMATTING:

March 17, 2022

RE: Life Science Alliance Manuscript #LSA-2022-01398-TR

Prof. Yutaka Suzuki
University of Tokyo
Department of Medical Genome Science
5-1-5 Kashiwanoha
Kashiwa, Chiba 277-8562
Japan

Dear Dr. Suzuki,

Thank you for submitting your revised manuscript entitled "Intensive Single Cell Analysis Reveals Immune Cell Diversity among Healthy Individuals". We would be happy to publish your paper in Life Science Alliance pending final revisions necessary to meet our formatting guidelines.

- please add the Twitter handle of your host institute/organization as well as your own or/and one of the authors in our system
- please check your callouts for your Tables. You have a callout for Table S15, but this is not in the legend or uploaded in the table file
- please add a panel C to your Supplementary Figure 8 legend and add a callout for Figure S8C in the main manuscript text
- due to online formatting requirements, LSA requires that the figure files do not exceed 1 page. We do not have a limit on the number of Figures used. Please adjust your Figures 1, 4, 6 & 7 accordingly, editing the figure callouts where necessary.

A. FINAL FILES:

B. MANUSCRIPT ORGANIZATION AND FORMATTING:

Sincerely,

March 22, 2022

RE: Life Science Alliance Manuscript #LSA-2022-01398-TRR

Prof. Yutaka Suzuki
University of Tokyo
Department of Medical Genome Science
5-1-5 Kashiwanoha
Kashiwa, Chiba 277-8562
Japan

Dear Dr. Suzuki,

Thank you for submitting your Resource entitled "Intensive Single Cell Analysis Reveals Immune Cell Diversity among Healthy Individuals". It is a pleasure to let you know that your manuscript is now accepted for publication in Life Science Alliance. Congratulations on this interesting work.

DISTRIBUTION OF MATERIALS:

Again, congratulations on a very nice paper. I hope you found the review process to be constructive and are pleased with how the manuscript was handled editorially. We look forward to future exciting submissions from your lab.

Sincerely,
